# Assessment of the functional state of the back muscles in girls with C-shaped low-grade scoliosis in a tensiomyographic image: An observational cross-sectional study

Tomasz Szurmik[1], Katarzyna Ogrodzka-Ciechanowicz [2]*, Piotr Kurzeja[3], Bartłomiej Gąsienica-Walczak[3], Jarosław Prusak[3], Karol Bibrowicz[4]

1 Faculty of Arts and Educational Science, University of Silesia, Cieszyn, Poland, 2 Institute of Clinical Rehabilitation, Faculty of Motor Rehabilitation, University of Physical Education, Krakow, Poland, 3 Institute of Health Sciences, Podhale State College of Applied Sciences, Nowy Targ, Poland, 4 Science and Research Center of Body Posture, Kazimiera Milanowska College of Education and Therapy, Poznań, Poland

* katarzyna.ogrodzka@awf.krakow.pl

**Data Availability Statement:** The minimal data set is contained within our paper.

## Abstract

### Purpose

The study aimed to test the feasibility of using tensiomyography to assess the functional status of the latissimus dorsi and erector spinae muscles in girls with C-shaped low back scoliosis.

### Materials and methods

Twenty-five girls aged 13–15 took part in an observational (cross-sectional) study. The examination involved measurements using the tensiomyography method (TMG). Two groups of muscles were tested: latissimus dorsi and erector spinae on the concave and convex side of low-grade scoliosis. The following indicators were analyzed: Td–delay time, Tc–contraction time, and Dm–maximal muscle displacement.

### Results

The analysis of Td revealed that values of this variable on the concave side were slightly lower compared to the convex side in both tested groups of muscles. Similarly, Tc values on the concave side were slightly lower than on the convex side of the curvature in both groups of muscles. In the case of Dm, lower displacement values and, consequently, greater muscle rigidity were observed on the concave side of the latissimus dorsi and the convex side of the erector spinae.

### Conclusions

The TMG method can be potentially used to diagnose the functional condition of muscles in patients with low-grade scoliosis. There were differences between the functional condition of the muscles on the concave and convex sides of the curvature.

**Funding:** The author(s) received no specific funding for this work.

**Competing interests:** The authors have declared that no competing interests exist.

## Introduction

Scoliosis is a three-plane deformation of the spine and also the trunk, which may result in negative self-image, pain, and potential negative effects connected with orthotic or surgical treatment [1].

The onset and development of scoliosis may depend on etiological and biomechanical factors. Etiological factors may be very diverse and initiate scoliosis. The biomechanical factor is, in turn, typical for all types of scoliosis and is subject to the laws of gravity and growth regardless of etiology. This factor controls the development of the deformation [2–5].

However, some believe that original disorders connected with the development of scoliosis which lead to muscle imbalance can be found in the central nervous system (CNS). Therefore, changes in the passive elements of the spine could be secondary [6]. In scoliosis, the muscle balance of the main spine stabilizers, or spinal erectors, may be distorted due to original dysfunctions of CNS structures [7]. This, in turn, may lead to disorders of erector spinae muscles' functioning and tension, causing abnormal curvatures of the spine. There are many arguments showing that discrete functional changes in the CNS are the actual cause of idiopathic scoliosis [8]. Not without significance is the issue of postural stability, which is responsible for maintaining the correct body position [9, 10].

In both diagnostics and prevention of scoliosis, there is a strong focus on the assessment of differences in the tension of the erector spinae muscles on the concave and convex sides of scoliosis [11].

Determination of asymmetries (differences) in erector spinae tensions between the concave and the convex side of the curvature seems to be essential when it comes to selecting the best conservative treatment method [12]

The muscular system is considered to be a coherent structure of co-dependent elements. There are many theories that explain the complex relations between certain muscle groups and individual muscles. The majority of the contemporary theses are based on the so-called functional muscular and myofascial chains [13].

Tensiomyography (TMG) is a method that enables precise measurement of contraction time and muscle stiffness, and the assessment of the changes in muscle belly displacement during an electrically stimulated isometric contraction response. This method of examination of muscle morphophysiology has been known for almost 20 years now and, compared to electromyography, is an easier method for obtaining information about the functioning of the muscle. It was developed in 1983 by Professor Vojko Valenčič. Unlike EMG, it does not provide information about the activity of certain muscles in the analyzed function but as a noninvasive method, it allows the detection of skeletal muscle properties. TMG assesses the changes in muscle belly displacement in millimeters (mm) and the duration expressed in milliseconds (ms) in response to a single electric stimulus [14–16].

TMG does not require any effort from the patient and therefore it is often used to assess the functions of muscles after physical activity [17].

In recent years, tensiomyography has become an important and highly reliable technique for assessing muscle contractile properties [18–20].

TMG is increasingly often used in sports medicine, especially to assess the neuromuscular properties of various muscles as well as functional efficiency and recovery after effort [21–23].

However, there are no reports on using TMG in the assessment of spinal muscle contractile properties in scoliosis.

The purpose of this study was to test the feasibility of using tensiomyography for assessing the functional status of the latissimus dorsi and erector spinae muscles in girls with C-shaped low back scoliosis.

It was hypothesized that tensiomyography may be an element of early scoliosis prevention and detection of changes in the function of the back muscles before the clinical symptoms of scoliosis become observable.

## Material and methods

### Study design

This is an observational cross-sectional study. The study protocol follows the guidelines of the Helsinki Declaration. This study was conducted in compliance with the Strengthening the Reporting of Observational Studies in Epidemiology (STROBE) Statement: guidelines for reporting observational studies [24]. The study was approved by the Bioethics Committee at the Kazimiera Milanowska College of Education and Therapy in Poznań, approval No. 003/ 2019 (approval date 15/02/2019). Informed consent was obtained from the parents/ legally authorized representatives of subjects who were under 16 at the moment of the study.

### Setting

The study was conducted in July 2021, during a rehabilitation holiday in Sarbinowo, organized by the Orto-Med clinic from Bielsko-Biała. Children with low-grade lumbar scoliosis diagnosed by an orthopedist according to SOSORT guidelines participated in a preventive and therapeutic stay [25].

### Participants

The first author had qualified 30 female patients of a physiotherapy clinic, who were beginning their low-grade scoliosis therapy. The first author was responsible for the randomization of the patients. Direct measurements were carried out under blinded conditions. The randomization method was simple random sampling. The person responsible for the tensiomyography examination conducted it on the right and left side without information about the medical diagnosis. The results were assigned as occurring on the concave or convex side in the next part of the analysis of the results conducted by another researcher.

### Eligibility criteria

1. Gender: female;

2. Low-grade lumbar adolescent idiopathic scoliosis diagnosed by an orthopedist, the mean curvature angle between 11˚ and 15˚ according to Cobb (Mean = 13.2˚ SD = 1.4); Risser $\leq$ 3 (based on physical examination and X-ray examination with determination of the Cobb angle);

3. Patients have had their first menstruation;

4. No coexisting diseases or injuries which could affect the test results (i.e. leg length discrepancy);

5. Written consent of a parent (guardian) for the patient to participate in the study.

6. All the participants declared that their physical and recreational activity is moderate and that they were not engaged in competitive sports.

7. Participants had not previously participated in scoliosis therapy.

## Outcome measures

The examination involved measurements using the tensiomyography method. Two groups of muscles were tested: latissimus dorsi (LD) and erector spinae (ES) on the concave and convex sides of low-grade scoliosis. The co-author responsible for direct tensomyographic measurements conducted them under single-blind conditions.

The variables assessed in this study were maximal muscle displacement (Dm), contraction time (Tc), and delay time (Td), as these parameters have been shown to have a low error rate (0.5% to 2.0%) and high reproducibility (ICC: 0.85–0.98) [26–29].

1. Td–delay time, determined between the electric impulse and 10% of the muscle response (ms)

2. Tc–contraction time, from 10% to 90% of the maximal response from the muscle, measured in milliseconds (ms)

3. Dm–maximal muscle displacement, measured in millimeters (mm)

The percentage of lateral symmetry (LS) was calculated using an algorithm implemented by the TMG-BMC tensiomyography software:

$$Ls = 0.1 \cdot \frac{\min(TDr \cdot TDl)}{\max(TDr \cdot TDl)} + 0.6 \cdot \frac{\min(TCr \cdot TCl)}{\max(TCr \cdot TCl)} + 0.1 \cdot \frac{\min(TSr \cdot TSl)}{\max(TSr \cdot TSl)} + 0.2$$
$$\cdot \frac{\min(DMr \cdot DMl)}{\max(DMr \cdot DMl)|}$$

## Assessment

The tests were carried out using a TMGTM Science for Body Evolution system (TMG-S2 produced by TMG-BMC d.o.o.). The TMG equipment consists of four components: impulse generating unit, electrode, mechanical sensor, and control unit with data recording and analysis software. The surface electrodes are attached to the skin, over the muscle belly, according to the direction of the muscle fibers, possibly halfway the muscle belly length, about 5 cm from one another. The head of the mechanical sensor is attached between the electrodes. The sensor is mounted on a stand which enables precise manipulation of the sensor's position and its stabilization during the measurement. The sensor measures the thickness of the belly stimulated by an electric impulse. During TMG, the changes in muscle belly displacement in millimeters (mm) and time expressed in milliseconds (ms) in response to a single electric stimulus are assessed. Each measurement tests several muscle response indicators. The most precise information can be obtained through the analysis of the contraction time (Tc)–from 10% to 90% of the maximal response from the muscle, measured in milliseconds (ms) and the maximal displacement (Dm), measured in millimeters (mm). Dm provides information about the tension of skeletal muscles as well as their morphological and structural changes. Tc refers to the time between 10% and 90% of Dm and is connected with the activity of the slow-twitch fibers. TMG also determines several other indicators, such as delay time (Td), determined between 0% and 10% of the maximal response, measured in milliseconds, sustain time (Ts), determined from 50% of the maximal response during return to the baseline condition, expressed in milliseconds, or relaxation time (Tr), measured in milliseconds which is determined after obtaining the maximum of Dm and taken to fall from 90% to 50% of the maximal response. The measurement data form a characteristic graph of the displacement in time. The values obtained are presented in Fig 1. In order to illustrate the characteristics of certain time variables, displacement (measured in millimeters) is expressed as a percentage [15, 16, 19, 30–32].

Two self-adhesive electrodes, 2–4 cm in diameter, were attached to the subjects' bodies. The electrodes were located on the concave and convex sides of the curvature at the level of the L3 process for the er. spinae and for the lat. dorsi according to the direction of muscle fibers with the use of TMG and SENIAM methodology [33].

Electrode diameter was selected based on the muscle size to isolate the contraction of a specific muscle and avoid activation of the neighboring muscles. A single one-phase square impulse of 1 ms duration was delivered from the electrostimulator to the electrode to evoke muscle contraction transdermally. The impulse power was gradually increased by 10 mA until a peak muscle contraction response was obtained. Typical maximal responses were observed between 40 and 90 mA. In order to minimize fatigue effects, 10-second intervals were maintained between each stimulation impulse. The area where the sensor was attached was selected according to the TMG scheme, in the thickest part of the muscle. If necessary, the area of sensor attachment was slightly corrected afterward to obtain the highest mechanical response. The sensor adhered to the skin half distance between the electrodes, about 5 cm from their centers.

The digital TMG signal was generated directly from the sensor using a sampling frequency of 1 kHz. After the measurement, the TMG signal data was stored on a PC drive. Maximum values obtained from two measurements were recorded and averaged for further analyses. The accepted maximal stimulation amplitude was the minimal amplitude needed for the response with the maximal muscle displacement (Dm).

The tests were performed on the first day of the rehabilitation holiday in a physician's office between 8 a.m. and 1 p.m. The examined person assumed the prone position with their arms along the body, the head in a position enabling relaxation of the whole body, and with rollers placed under the talocrural joints. The electrodes were attached along the muscle bellies of the latissimus dorsi and erector spinae on both sides of the curvature (concave and convex sides). The contraction time was evoked by single electric stimuli. Self-adhesive electrodes were placed around the TMG sensor. The anode was attached distally and the cathode–proximally, 20–50 mm from the measurement point. Bipolar electric stimulation involved a single direct current impulse that lasted one millisecond.

## Statistical methods

The statistical analysis of the results was conducted using a MedCalc ver. 20.015 package.

The results were analyzed and described by the authors of the publication in the conditions of a blinded trial. The sample size was not specified, the study was conducted on all participants of the rehabilitation stay who met the criteria for inclusion in the study. The characteristics of the distribution of variables were determined using the Shapiro–Wilk method. Mean values and standard deviation were calculated. As the distribution of the variables was normal, the analysis of bilateral variables was conducted using Student's t-test for independent samples. Differences were considered significant at $p < 0.05$.

## Results

The sample consisted of 30 girls. Due to the written withdrawal of the parent/guardian consent for a child's participation in the study and ineligibility, 25 girls aged 13–15, whose morphological parameters were typical for their age, were qualified for the final stage of the tests. Table 1. presents the detailed anthropometric data of the sample. Fig 2. shows the qualification stage.

The analysis of the delay time in response to electric stimulus (Td) shows similar trends observed in both examined muscles. The values on the concave side were slightly lower than on the convex side of the curvature. It was also observed that the contraction times of ES were

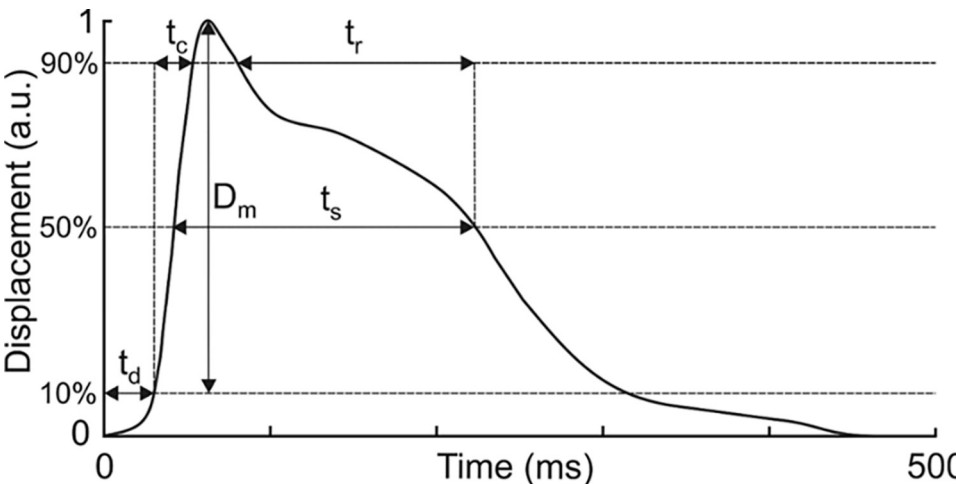

**Fig 1. TMG displacement and the measured indicators. Delay time (Td)** as a time between the electrical impulse and 10% of the contraction; **Contraction time (Tc)** as a time between 10% and 90% of the contraction; **Sustain time (Ts)** as a time between 50% of the contraction and 50% of the relaxation; **Relaxation time (Tr)** as a time between 90% and 50% of the relaxation; **Displacement (Dm)** as maximal amplitude of the muscle contraction (https://www.tmg-bodyevolution.com/research/tmg-research-measuring-device/).

shorter compared to LD (convex side p = 0.134, concave side p = 0.048*). Only the differences for the concave side were statistically significant. The bilateral analysis of the muscles indicates similar differences. Delay time values on the convex side were slightly higher than on the convex side in both examined muscles (LD convex side X = 24.8/SD = 7.29 concave side x = 24.3/SD = 8.04. ES convex side X = 22.3/SD = 3.29, concave side X = 20.9/SD = 2.39). However, the differences recorded on the convex and concave sides were not statistically significant (LD p = 0.843, ES p = 0.092). (Table 2).

The analysis of the contraction time (Tc) shows similar trends observed in both examined muscles. The values on the concave side were slightly lower than on the convex side of the curvature. It was also observed that contraction times of ES were significantly shorter compared to LD (convex side p = 0.039. concave side p = 0.021). The bilateral analysis of the muscles indicates similar differentiation. Contraction time values on the convex side were slightly higher than on the convex side in both examined muscles (LD convex side X = 18.9/SD = 5.74. concave side x = 18.6/SD = 4.26. ES convex side X = 16.3/SD = 2.38. concave side X = 16.2/SD = 2.54). However. the differences recorded on the convex and concave sides were not statistically significant (LD p = 0.928. ES p = 0.945). (Table 3).

Examination of maximal displacement (Dm) does not reveal such tendencies as the measurements of contraction time. Lower displacement values and. consequently. greater muscle stiffness were observed on the concave side of LD and the convex side of ES. The detailed bilateral analysis of the muscles indicates similar differences (LD: convex side X = 1.2/SD = 0.93.

**Table 1. Study group.**

| Variable | X | SD | min–max |
|---|---|---|---|
| Age [yrs] | 14.1 | 1.25 | 13–15 |
| Body weight [kg] | 42.6 | 9.75 | 35–58 |
| Height [cm] | 153.6 | 11.24 | 141–161 |

X–mean, SD–standard deviation

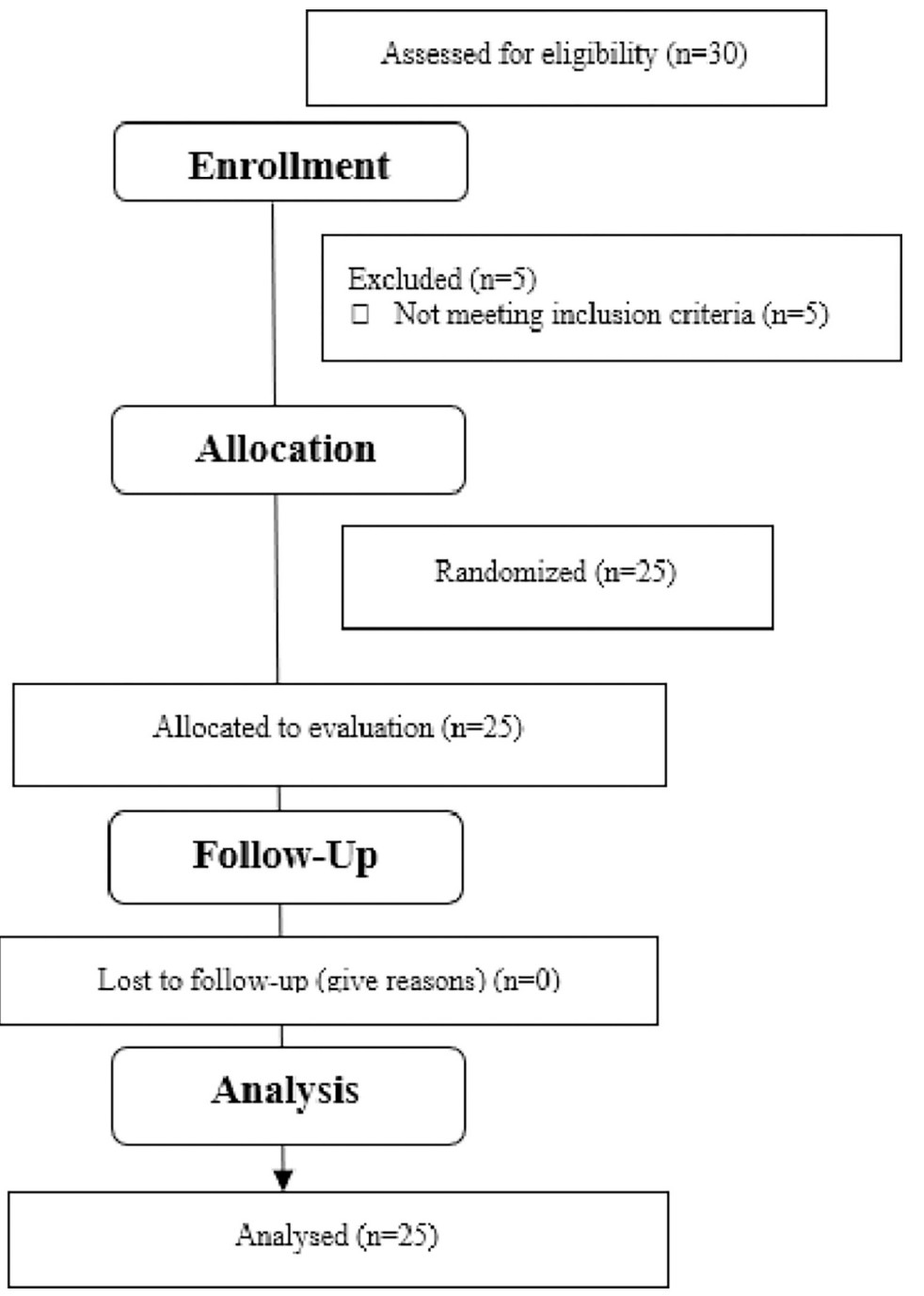

**Fig 2. Flow diagram.**

concave side X = 09/SD = 0.63; ES: convex side X = 2.35/SD = 1.33. concave side X = 2.43/SD = 142). However. for both examined muscles. the differences in displacement on the convex and concave sides were not statistically significant (LD: p = 0.151; ES: p = 0.823). There were. however. clear differences between the values of this variable for the LD and ES (convex side p = 0.0009. concave side p<0.0001). (Table 4).

**Table 2. Mean values, standard deviations and results of difference analysis for Td (Delay time).**

| Muscle | Td [ms] | | p |
|:---:|:---:|:---:|:---:|
| | Concave side X±SD | Convex side X±SD | |
| ES | 20.9±2.39 | 22.3±3.29 | 0.092 |
| LD | 24.3±8.04 | 24.8±7.29 | 0.843 |
| p | 0.048* | 0.134 | |

*- statistically significant

**Table 3. Mean values, standard deviations and results of difference analysis for Tc (Contraction time).**

| Muscle | Tc [ms] | | p |
|:---:|:---:|:---:|:---:|
| | Concave side X±SD | Convex side X±SD | |
| ES | 16.2±2.54 | 16.3±2.38 | 0.945 |
| LD | 18.6±4.26 | 18.9±5.74 | 0.928 |
| p | 0.021* | 0.039* | |

*- statistically significant

**Table 4. Mean values, standard deviations and results of difference analysis for Dm (Displacement).**

| Muscle | Dm [mm] | | p |
|:---:|:---:|:---:|:---:|
| | Concave side X±SD | Concave side X±SD | |
| ES | 2.4±2.54 | 2.5±1.42 | 0.945 |
| LD | 0.9±0.63 | 1.2±0.93 | 0.928 |
| p | <0.0001* | 0.0009* | |

*- statistically significant
The bilateral symmetry index for LD muscles was 65.1% and 77.9% for ES muscles.

## Discussion

The study aimed to analyze the possible application of TMG in the early detection of functional changes in back muscles in patients with low-grade scoliosis.

During the preparation of this study. the authors assumed the possibility of early detection of changes in the biomechanical function of the back muscles. even preceding the occurrence of clinical symptoms of scoliosis. This could contribute to earlier detection of the risk of scoliosis and early treatment at the pre-scoliotic stage.

The analysis of the delay time (Td) in response to electric impulse revealed that values of this variable on the concave side were slightly lower compared to the convex side in both tested groups of muscles. Similarly. Tc (contraction time) values on the concave side were slightly lower than on the convex side of the curvature in both groups of muscles. In the case of Dm (maximal displacement). lower displacement values and. consequently. greater muscle rigidity were observed on the concave side of the latissimus dorsi and the convex side of the erector spinae.

The examination of bilateral asymmetry for both ES and LD clearly indicates its existence. The asymmetry was more noticeable for the LD muscles. This could indicate that LD muscles as phasic muscles may be a better predictor of scoliosis compared to paraspinal muscles. which are postural muscles.

The assessment of the mechanical properties of muscles can be an additional diagnostic element for assessing the adaptive processes of skeletal muscles (for example. hypertrophy. muscle strength). Using TMG. in particular for the interpretation of two indicators: contraction time (Tc) and displacement (Dm). may support such assessment. Thus. TMG becomes an important and effective method of measuring muscle contractile properties (MCP) [18].

TMG enables quick determination of MCP in response to muscle function [34]. It seems that muscle contractile parameters obtained from TMG may have significance in the noninvasive assessment of muscle quality in population studies [34]. Some reports confirm the possibility of the application of TMG in the assessment of changes in certain groups of muscles and lateral asymmetries [35, 36]. As for TMG indicators. some studies show that Dm and Tc are the most reliable ones [27, 29]. TMG may be used to measure the properties related to muscle contractility and mechanical responses based on the displacement of the muscle belly [37–39].

Assessment of Dm which is susceptible to resistance training (RT) stimulus suggests that RT results in lower Dm values (that is. greater rigidity) in both acute and chronic conditions [27, 39]. The authors' original tests revealed that lower displacement values and. consequently. greater muscle rigidity were observed on the concave side of the latissimus dorsi and the convex side of the erector spinae. Also. greater rigidity of the latissimus dorsi compared to the erector spinae was observed. It is interesting because the latissimus dorsi is classified rather as a phasic muscle whereas the erector spinae is a typical tonic muscle. Different Dm values may be associated with adaptive morphological changes in the muscles. thus. Dm may be used to assess the effectiveness of RT in the case of muscle hypertrophy [40]. Low maximal displacement values usually indicate greater muscle tension [40]. Higher Dm values are associated with reduced physical activity [30, 41]. According to the authors. confirmation of the increased rigidity of the latissimus dorsi on the curvature's concave side can be an important diagnostic and prognostic hint in the perspective of pathomechanical changes which may affect further progression of scoliosis. Increased Dm may be connected with the atrophy of muscle fibers. the adaptation of muscle architecture towards reducing their thickness. and the length of muscle fibers [30]. Growing Dm values may also indicate decreased rigidity [41]. Therefore. it may be important to consider the rigidity of certain groups of back muscles during planning the conservative treatment (physiotherapy) of scolioses. It seems. however. that the exact mechanisms underlying the decrease in Dm. for example during fatigue. remain unexplained. It must also be pointed out that changes that occur in the assessment of Dm are harder to interpret. probably due to difficulty in the determination of muscle rigidity in vivo [42].

Both time parameters (Tc and Td) are temporal parameters relating to the time of the muscle response to an electrical impulse. Their course had similar characteristics on both concave and convex sides of the curvature. Tc can be connected with certain types of muscular fibers. where lower Tc values are correlated with slow-twitch fibers [32]. Higher Tc values may suggest greater involvement of type I fibers characterized by weaker and slower twitch [43]. The analysis of contraction time (Tc) reveals similar trends observed in both examined muscles (erector spinae. latissimus dorsi). It was also observed that contraction times of the erector spinae were significantly shorter compared to the latissimus dorsi. The authors also noted that Tc values on the convex side were slightly higher than on the convex side in both examined muscles. It might be connected with a developing mechanism of stretching this group of muscles. Perhaps. such interpretation should also be considered in scoliosis diagnostics and therapy. especially from the perspective of potential progression resulting from the natural history of the development of scoliosis. Mannion et al. observed that in individuals without a diagnosis of abnormal curvature. type I muscle fibers were distributed symmetrically. Individuals with scoliosis had a lower amount of fibers of this type on the concave side of the curvature [44]. The question that should be asked is. does this situation occur only in advanced scolioses? The

results of the authors' study show that the differences in response time to an electric impulse can be found already in low-grade curvatures. which would confirm the observations made by Mannion et al. [44].

According to Rusu et al. shorter Tc is connected with greater involvement of type II. fast-twitch fibers [45]. Haff et al. note that type II fibers are the main fibers responsible for generating maximum strength and fast movements [46]. The growth of Tc values is also observed during muscle fatigue [47]. Some authors suggest that higher Tc and lower Dm values may be associated with greater strength-related effort of certain groups of muscles [48–50]. The Cobb angle correlates with the greater involvement of type I fibers on the convex side of the spine curvature [51]. Trontelj and Fernandez examined the spinal reflexes (stretch reflexes) in patients with scoliosis and concluded that the response to the phasal stretch of the paravertebral muscles is symmetrical in all the examined individuals [52]. The authors formulated a hypothesis that regional neurogenic disorders may play an important role in the development of scoliosis. These disorders may occur mainly in deep paravertebral muscles on the convex side of the curvature.

Already in 1983. Sahgal et al. performed a morphological test of paravertebral muscle samples obtained from a biopsy and observed changes in the types of muscular fibers in patients with adolescent idiopathic scoliosis (AIS) [53]. The authors suggested that the causes of AIS can be found in muscle etiology. The authors' original study shows that the asymmetry of contractile properties of the back muscles is noticeable already in the early stages of scoliosis.

Some authors suggest that loads imposed on the paravertebral muscles on the concave and convex sides of the curvature are proportional to their functional efficiency [54]. In this way. muscles on the convex side are adapted to accept greater loads. These observations were confirmed by the results of paravertebral muscle morphology tests.

According to the authors' knowledge. it is the first study that used TMG to assess the muscles on the concave and convex sides of the curvature in patients with scoliosis.

There are no clear research that would make it possible to determine whether paravertebral muscles have a direct impact on the development or progression of AIS. It is also not clear whether their increased activity is to prevent the progression of scoliosis or whether it changes in certain stages of the disease.

The pathogenesis of AIS remains unclear. but most likely it is multifactorial. Factors including greater height. delayed puberty and late menarche in females. and low BMI have been shown to contribute to it [55–57]. Although asymmetry in muscle activity could not be verified as the cause or result of scoliosis. extant literature asserts that muscle asymmetry of ES in patients with AIS promotes its development [51, 58]. Some authors found signs of dystrophy and atrophy in the back muscles and differences in proportions of type I fibers versus type II fibers with greater of the former on the apex of the convex side [59, 60]. Not only reduced proportion of type I fibers on the concave side but also a decrease in total muscle volume was observed [61].

Given the observation of the difference in muscle tension potentials in individual muscle groups. the authors tried to look for their relationship to the development in the early stage of AIS. Deficits in paraspinal muscle strength and endurance due to changes in muscle composition in patients with AIS potentially compromise their ability to maintain postural stability. which may be reflected in spinal kinematics.

Delineating the underlying multifactorial pathogenesis of AIS could pave the way to the identification of those at risk of both its onset and progression. to identify which individuals need closer monitoring.

The etiology of scoliosis might be multifactorial including paraspinal asymmetry. so the presented study explored the factors which contribute to the Cobb angle. Analysis of these parameters

could provide more information regarding the understanding of AIS. Subsequently. this information could be used to develop more precise. science-based exercise treatment methods.

The authors' original study suggests that TMG can be very helpful in clinical tests: for identifying the characteristics which appear at different moments and abnormalities in spine growth in patients with scoliosis. as well as for quantitatively determining correct values. In other words. verification and identification of the properties of various abnormal erector spinae by means of TMG may be of great importance in the initial phase of idiopathic scoliosis. when conservative treatment methods are proposed. In the future. patient assessments should be multidimensional and include an initial assessment of muscle strength and/or endurance. determination of the mechanisms of physiological adaptation of the back muscles in scoliosis. assessment of muscle fatigue. and further application of TMG to measure dynamic muscle contractions. The authors of the paper hope that this study will serve as a basis for further tests using TMG in patients with idiopathic scoliosis. As the reliability and accuracy of TMG in patients with scoliosis have not been described yet. before using TMG in this group of patients further studies should be conducted to determine its effectiveness.

Furthermore the assessment of contractile properties of skeletal muscles with the use of TMG may provide insights into the pace of changes in stimulation-contraction feedback in relation to architectural adaptations. Importantly. TMG may not only provide information about the muscle architecture but may also become an objective and simple method of assessing skeletal muscles in their "active" state as compared to. for example. an assessment carried out at rest. for instance during ultrasonography.

## Study limitations

There are some limitations to this study. There were no comparisons with a control group. As muscles with long lever arms have higher deformational forces. follow-up studies evaluating both paraspinal and large muscles are required. This study was a cross-sectional physiologic study so it could not identify any causal relationships. A future prospective longitudinal study to evaluate scoliosis causality is required.

The study reveals that TMG is useful for assessing the pathomechanism and physiotherapy of scoliosis but more research is needed to develop premises and full recommendations regarding the use of this method in tracking changes in the muscular system caused by scoliosis. So far. there has been no information about using TMG in the diagnostics of scoliosis and evaluation of treatment results.

## Conclusions

1. The TMG method can be potentially used to diagnose the functional condition of muscles in patients with low-grade scoliosis.

2. There were differences between the functional condition of the muscles on the concave and convex sides of the curvature.

3. The study did not show clear differences in the functional condition of the erector spinae and latissimus dorsi among the examined patients.

## Supporting information

**S1 Checklist. STROBE statement—checklist of items that should be included in reports of *cross-sectional studies*.**
(DOC)

**S1 Protocol.**
(DOCX)

## Author Contributions

**Conceptualization:** Tomasz Szurmik, Katarzyna Ogrodzka-Ciechanowicz, Piotr Kurzeja, Karol Bibrowicz.

**Data curation:** Tomasz Szurmik.

**Formal analysis:** Tomasz Szurmik, Karol Bibrowicz.

**Investigation:** Tomasz Szurmik, Karol Bibrowicz.

**Methodology:** Tomasz Szurmik, Katarzyna Ogrodzka-Ciechanowicz, Piotr Kurzeja, Karol Bibrowicz.

**Supervision:** Katarzyna Ogrodzka-Ciechanowicz.

**Visualization:** Katarzyna Ogrodzka-Ciechanowicz, Piotr Kurzeja, Bartłomiej Gąsienica-Walczak, Jarosław Prusak.

**Writing – original draft:** Tomasz Szurmik, Piotr Kurzeja, Karol Bibrowicz.

**Writing – review & editing:** Katarzyna Ogrodzka-Ciechanowicz, Bartłomiej Gąsienica-Walczak, Jarosław Prusak, Karol Bibrowicz.

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
