## [Decision Letter · Decision Letter 0]

29 Jun 2023

PONE-D-23-14959Assessment of the functional state of the back muscles in girls with C-shape low-grade scoliosis in a tensiomiographic image: an observational cross-sectional studyPLOS ONE

Dear Dr. Ogrodzka-Ciechanowicz,

Thank you for submitting your manuscript to PLOS ONE. After careful consideration, we feel that it has merit but does not fully meet PLOS ONE’s publication criteria as it currently stands. Therefore, we invite you to submit a revised version of the manuscript that addresses the points raised during the review process.

We look forward to receiving your revised manuscript.

Kind regards,

Monika Błaszczyszyn

Academic Editor

PLOS ONE

Journal Requirements:

Reviewers' comments:

Reviewer's Responses to Questions

**Comments to the Author**

1. Is the manuscript technically sound, and do the data support the conclusions?

Reviewer #1: Yes

Reviewer #2: Partly

2. Has the statistical analysis been performed appropriately and rigorously? 

Reviewer #1: Yes

Reviewer #2: No

3. Have the authors made all data underlying the findings in their manuscript fully available?

Reviewer #1: Yes

Reviewer #2: Yes

4. Is the manuscript presented in an intelligible fashion and written in standard English?

Reviewer #1: Yes

Reviewer #2: No

5. Review Comments to the Author

Reviewer #1: The manuscript provided by the author contains data analysis on different muscle activity parameters of Erector Spinae and Latissimus Dorsi on the convex and concave side of the spine. The author has performed two types of comparisons: (1) between the two muscle groups on the same side of the spine, and (2) between the two sides of the spine for each muscle group.

However, the author should clarify which type of comparison they are referring to in each sentence, and use consistent terminology to avoid confusion. For example, the author could say "the value of Erector Spinae on the concave side was compared with the value of Latissimus Dorsi on the concave side" instead of "the value of the concave side of Erector Spinae and Latissimus Doris".

The author has drawn a clear conclusion from the data, and has acknowledged the limitation of the small sample size. The manuscript is well-written and demonstrates a rigorous methodology. The author could also consider other methods to measure muscle activity, such as EMG[1], MRI[2], or Ultrasound[3], and discuss their advantages and disadvantages compared to the method used in this study.

[1]Garcia Kwok, Joanne Yip, Mei-Chun Cheung, Kit-Lun Yick,

"Evaluation of Myoelectric Activity of Paraspinal Muscles in Adolescents with Idiopathic Scoliosis during Habitual Standing and Sitting",

https://doi.org/10.1155/2015/958450

[2]Gnahoua Zoabli, Pierre A. Mathieu, Carl-Éric Aubin,

@Back muscles biometry in adolescent idiopathic scoliosis@,

https://doi.org/10.1016/j.spinee.2006.04.001.

[3]Borna S, Noormohammadpour P, Linek P, Mansournia M A, Kordi R.

"Ultrasound Measurements of the Lateral Abdominal Muscle Thicknesses in Girls With Adolescent Idiopathic Scoliosis",

https://doi.org/10.5812/asjsm.32274.

Reviewer #2: Skoliose

This observational study evaluates the muscle mechanical characteristics of two muscles (i.e., bilateral Erector spinae and Latissimus dorsi) in female participants with C Curve scoliosis. The authors measured these characteristics by use of tensiomyography (TMG) - a non-invasive technique which assesses the radial muscle displacement with a contact sensor following electrically evoked stimulation of the skeletal muscle of interest at rest. This is an interesting and novel study, and the topic covers a present gap in TMG literature which has been well recognised by the authors.

In the following, I address some critical points in terms of substantive references to the measurement procedure, statistics, and interpretation of the results.

Firstly, the manuscript should be completely revised linguistically. Partly, the sentence structure/grammar is not correct, making the text difficult to read.

Furthermore, I strongly recommend the authors to look into the literature and the procedure of the TMG. Unfortunately, there are numerous incorrect statements about the method throughout the manuscript.

Introduction:

Overall, the introduction should be significantly shortened. There are numerous repetitions. The authors should formulate a clear hypothesis at the end.

Page 3, lines 49-53: The statements of the two sentences are redundant, and therefore should be revised.

Page 3, lines 54-58: This section is partly a repetition of the previous one. Both paragraphs should be summarized in condensed form.

Page 4, lines 74-77: In this context, I suggest mentioning myofascial chains, instead of tendon chains.

Please see: doi: 10.1016/j.apmr.2015.07.023

Page 4, line 79: TMG measures time of contraction and not the speed. The velocity of contraction is a secondary parameter which can be calculated by combining, for example, Dm and Tc and/or Td, please see:

doi: 10.1371/journal.pone.0262156 and DOI: 10.1519/JSC.0000000000004495

Page 4, line 81: TMG does not measure the morphology of a muscle, such as Ultrasound or Elastography.

Page 4, line 82: TMG and EMG are two entirely different procedures. TMG measures the mechanical characteristics of a muscle, whereas EMG measures the electrical activity of the muscle.

Page 5, line 95: This is not correct. By now, there are numerous publications on TMG and spinal/trunk muscles:

doi.org/10.1007/s11332-023-01083-7

DOI: 10.3390/jcm11113198

doi: 10.1016/j.clinbiomech.2021.105351

doi: 10.1177/1941738120917932

doi: 10.1016/j.jbiomech.2019.05.035

doi: 10.1590/1806-9282.64.06.549

doi: 10.1007/s00421-018-3867-2

Considering the aspect of TMG and Scoliosis, the authors are right.

Last paragraph of the introduction: What is the authors’ hypothesis?

Methods:

Page 6, line 106 and 134: What do the authors mean by “blinded conditions” in terms of TMG assessment?

Page 6, line 132: I suggest using more common abbreviations: ES and LS

Methods – Assessment: What was the authors’ rational to implement Td, Tc, and Dm?

What is the difference between Td and Tc? And I wonder why the secondary parameter Vc/Vr was not mentioned?

Moreover, the TMG software provides information about the lateral symmetry of bilateral muscles (Ls). I recommend implementing this parameter as it provides further information about muscle-mechanical function.

doi.org/10.1007/s11332-023-01083-7

doi: 10.1016/j.clinbiomech.2021.105351

Page 7, line 149: TMG does not measure the thickest part of the muscle. The assessor needs to evaluate the thickest part of the muscle in order to obtain the maximal muscle response.

Statistical methods:

This section needs to be revised, and the effect size should be mentioned. The authors have to counteract the problem of multiple comparisons.

Results:

Has to be revised, and I suggest showing the descriptive statistics in tabular form.

Page 10, line 217 ff: What does “clear tendency” mean in terms of significant results? A clear trend suggests something that is statistically not the case. This type of formulation is found throughout the manuscript and should be avoided.

Discussion:

The results do not support the authors' conclusions.

Were the subjects asked to avoid exercise prior to the study? If not, this should be included in the limitations.

6. PLOS authors have the option to publish the peer review history of their article (what does this mean?). If published, this will include your full peer review and any attached files.

Reviewer #1: No

Reviewer #2: No

---

## [Author Response · Author response to Decision Letter 0]

30 Aug 2023

Reviewer #1: 

The manuscript provided by the author contains data analysis on different muscle activity parameters of Erector Spinae and Latissimus Dorsi on the convex and concave side of the spine. The author has performed two types of comparisons: (1) between the two muscle groups on the same side of the spine, and (2) between the two sides of the spine for each muscle group.

1. However, the author should clarify which type of comparison they are referring to in each sentence, and use consistent terminology to avoid confusion. For example, the author could say "the value of Erector Spinae on the concave side was compared with the value of Latissimus Dorsi on the concave side" instead of "the value of the concave side of Erector Spinae and Latissimus Doris".

Response: The manuscript has been reviewed by a native speaker to improve reading fluency.

2. The author has drawn a clear conclusion from the data, and has acknowledged the limitation of the small sample size. The manuscript is well-written and demonstrates a rigorous methodology. The author could also consider other methods to measure muscle activity, such as EMG[1], MRI[2], or Ultrasound[3], and discuss their advantages and disadvantages compared to the method used in this study.

[1]Garcia Kwok, Joanne Yip, Mei-Chun Cheung, Kit-Lun Yick, "Evaluation of Myoelectric Activity of Paraspinal Muscles in Adolescents with Idiopathic Scoliosis during Habitual Standing and Sitting", https://doi.org/10.1155/2015/958450

[2]Gnahoua Zoabli, Pierre A. Mathieu, Carl-Éric Aubin. “Back muscles biometry in adolescent idiopathic scoliosis” https://doi.org/10.1016/j.spinee.2006.04.001.

[3]Borna S, Noormohammadpour P, Linek P, Mansournia M A, Kordi R. "Ultrasound Measurements of the Lateral Abdominal Muscle Thicknesses in Girls With Adolescent Idiopathic Scoliosis", https://doi.org/10.5812/asjsm.32274.

Response: TMG and EMG, MRI or USG are entirely different procedures. TMG measures the biomechanical characteristics of a muscle, whereas EMG measures the electrical activity of the muscle. Muscle USG examination evaluates the morphology of the muscle, e.g. by assessing its damage, and in turn MRI gives a very detailed picture of the musculoskeletal system, especially the spine, but will not evaluate the biomechanics of the muscle. Therefore, comparing the results of all these methods would not be possible to substantiate the hypothesis. None of these methods, with the exception of TMG, can be used to screen for scoliosis.

Reviewer #2: 

This observational study evaluates the muscle mechanical characteristics of two muscles (i.e., bilateral Erector spinae and Latissimus dorsi) in female participants with C Curve scoliosis. The authors measured these characteristics by use of tensiomyography (TMG) – a non-invasive technique which assesses the radial muscle displacement with a contact sensor following electrically evoked stimulation of the skeletal muscle of interest at rest. This is an interesting and novel study, and the topic covers a present gap in TMG literature which has been well recognised by the authors.

In the following, I address some critical points in terms of substantive references to the measurement procedure, statistics, and interpretation of the results.

Firstly, the manuscript should be completely revised linguistically. Partly, the sentence structure/grammar is not correct, making the text difficult to read.

Furthermore, I strongly recommend the authors to look into the literature and the procedure of the TMG. Unfortunately, there are numerous incorrect statements about the method throughout the manuscript.

Introduction:

Overall, the introduction should be significantly shortened. There are numerous repetitions. The authors should formulate a clear hypothesis at the end.

1. Page 3, lines 49-53: The statements of the two sentences are redundant, and therefore should be revised.

Response: according to the reviewer's comment, the paragraph was removed.

2. Page 3, lines 54-58: This section is partly a repetition of the previous one. Both paragraphs should be summarized in condensed form.

Response: thank you very much for your insightful suggestion. After deleting the previous paragraph, the text was redacted.

3. Page 4, lines 74-77: In this context, I suggest mentioning myofascial chains, instead of tendon chains. Please see: doi: 10.1016/j.apmr.2015.07.023

Response: of course the term was changed to myofascial chains.

4. Page 4, line 79: TMG measures time of contraction and not the speed. The velocity of contraction is a secondary parameter which can be calculated by combining, for example, Dm and Tc and/or Td, please see: doi: 10.1371/journal.pone.0262156 and DOI: 10.1519/JSC.0000000000004495

Response: The authors fully agree with the comment of the reviewer, inaccuracies result from imprecise translation. The text only mentions the contraction time and muscle stiffness. This has been changed in the text.

5. Page 4, line 81: TMG does not measure the morphology of a muscle, such as Ultrasound or Elastography.

Response: The authors fully agree with the reviewer's comment that TMGs do not measure muscle morphology like ultrasonography or elastography. The obtained results focus on the biomechanical aspects of muscle work. However, there is a possibility to assess the morphology of the muscle in the context of assessing the type of muscle fibers of the tested muscle.

6. Page 4, line 82: TMG and EMG are two entirely different procedures. TMG measures the mechanical characteristics of a muscle, whereas EMG measures the electrical activity of the muscle.

Response: The authors agree with the opinion of the reviewer that electromyography and tensiomyography are two different methods and they analyse separate aspects of muscle work. However, it is hard to deny that they can be used to assess muscle function.

7. Page 5, line 95: This is not correct. By now, there are numerous publications on TMG and spinal/trunk muscles:

doi.org/10.1007/s11332-023-01083-7

DOI: 10.3390/jcm11113198

doi: 10.1016/j.clinbiomech.2021.105351

doi: 10.1177/1941738120917932

doi: 10.1016/j.jbiomech.2019.05.035

doi: 10.1590/1806-9282.64.06.549

doi: 10.1007/s00421-018-3867-2

Considering the aspect of TMG and Scoliosis, the authors are right.

Response: Therefore, we propose the following wording: However, there are no reports on using TMG in the assessment of spinal muscle contractile property in scoliosis

8. Last paragraph of the introduction: What is the authors’ hypothesis?

Response: Changes made: The purpose of this study was to test the feasibility of using tensiomyography to assess the functional status of the Latissimus Dorsi and Erector Spinae muscles in girls with C-shaped low back scoliosis.

It was hypothesized that tensiomyography may be an element of early scoliosis prevention and detection of changes in the function of the back muscles before the clinical symptoms of scoliosis become observable.

Methods:

9. Page 6, line 106 and 134: What do the authors mean by “blinded conditions” in terms of TMG assessment?

Response: The person conducting the tensiomyography examination conducted it on the right and left side without information about the medical diagnosis. Assignment of the results as occurring on the concave or convex side took place in the next part of the analysis of the results conducted by another researcher. This information is included in the manuscript

10. Page 6, line 132: I suggest using more common abbreviations: ES and LD

Response: changed as suggested

11. Methods – Assessment: What was the authors’ rational to implement Td, Tc, and Dm?

What is the difference between Td and Tc? And I wonder why the secondary parameter Vc/Vr was not mentioned?

Response: According to the authors, the choice of parameters was intentional. Two time parameters Tc and Td were selected. They describe two different time intervals of muscle contraction Tc – refers to the time between 10 and 90% of the maximum muscle response measured in milliseconds (ms) and Td is defined as the time between 0 and 10% of the muscle response. They are therefore used to describe other functional characteristics. Dm, on the other hand, according to many reports best describes muscle stiffness. All these selected parameters are described as the best verified, repeatable and most reliable of the variables obtained in the tensiomyography study. Derived parameters, such as contraction velocity Vc, are mainly used in sports medicine, and their usefulness and methodology for calculating their value are still being analysed.

12. Moreover, the TMG software provides information about the lateral symmetry of bilateral muscles (Ls). I recommend implementing this parameter as it provides further information about muscle-mechanical function.

doi.org/10.1007/s11332-023-01083-7

doi: 10.1016/j.clinbiomech.2021.105351

Response: The reviewer is fully correct in the fact that such variables as indicators of bilateral and functional asymmetry, which are percentage indicators, are provided by the TMG software. These are complex parameters and are mainly used to analyze players in the context of predicting injury risks.

13. Page 7, line 149: TMG does not measure the thickest part of the muscle. The assessor needs to evaluate the thickest part of the muscle in order to obtain the maximal muscle response.

Response: Of course, TMG does not measure muscle thickness. It only measures the response of the muscle belly in the place selected by the diagnostician as the place of the largest cross-section subjected to the muscle impulse, and this is also presented in the text.

Statistical methods:

14. This section needs to be revised, and the effect size should be mentioned. The authors have to counteract the problem of multiple comparisons. 

Response: The selection of tests resulted from the characteristics of variable distributions, therefore parametric tests and comparisons of only two muscle groups were used as independent variables.

Results:

15. Has to be revised, and I suggest showing the descriptive statistics in tabular form. 

Response: Charts have been converted to tables.

16. Page 10, line 217: What does “clear tendency” mean in terms of significant results? A clear trend suggests something that is statistically not the case. This type of formulation is found throughout the manuscript and should be avoided.

Response: Although there were no statistically significant differences, some clinical changes in muscle asymmetry were visible, therefore, from the point of view of prevention and the use of TMG in the early assessment of patients, it is important to capture and highlight such changes. As suggested by the reviewer, the word "clear" was removed and the word "tendency" was left as an emphasis on the phenomenon.

Discussion:

17. The results do not support the authors' conclusions. Were the subjects asked to avoid exercise prior to the study? If not, this should be included in the limitations.

Response: Thank you for the thorough analysis, we hope that now everything will be clear. Information was added in the inclusion criteria that the children had not previously participated in any scoliosis therapy. A fragment describing the rationale for the topic was added to the discussion.

---

## [Decision Letter · Decision Letter 1]

25 Sep 2023

Assessment of the functional state of the back muscles in girls with C-shaped low-grade scoliosis in a tensiomyographic image:an observational cross-sectional study

PONE-D-23-14959R1

Dear Dr. Ogrodzka-Ciechanowicz,

We’re pleased to inform you that your manuscript has been judged scientifically suitable for publication and will be formally accepted for publication once it meets all outstanding technical requirements.

Kind regards,

Monika Błaszczyszyn

Academic Editor

PLOS ONE

Additional Editor Comments (optional):

Reviewers' comments:

Reviewer's Responses to Questions

**Comments to the Author**

1. If the authors have adequately addressed your comments raised in a previous round of review and you feel that this manuscript is now acceptable for publication, you may indicate that here to bypass the “Comments to the Author” section, enter your conflict of interest statement in the “Confidential to Editor” section, and submit your "Accept" recommendation.

Reviewer #1: All comments have been addressed

Reviewer #2: All comments have been addressed

2. Is the manuscript technically sound, and do the data support the conclusions?

Reviewer #1: Yes

Reviewer #2: Yes

3. Has the statistical analysis been performed appropriately and rigorously? 

Reviewer #1: Yes

Reviewer #2: Yes

4. Have the authors made all data underlying the findings in their manuscript fully available?

Reviewer #1: Yes

Reviewer #2: Yes

5. Is the manuscript presented in an intelligible fashion and written in standard English?

Reviewer #1: Yes

Reviewer #2: Yes

6. Review Comments to the Author

Reviewer #1: (No Response)

Reviewer #2: The authors are to be commended for the thorough revision of their manuscript. The paper has significantly improved as a result.

I have no further comments. Thank you very much.

7. PLOS authors have the option to publish the peer review history of their article (what does this mean?). If published, this will include your full peer review and any attached files.

Reviewer #1: **Yes: **Joanne Yip

Reviewer #2: No

---

## [Editor Report · Acceptance letter]

6 Oct 2023

PONE-D-23-14959R1 

Assessment of the functional state of the back muscles in girls with C-shaped low-grade scoliosis in a tensiomyographic image: an observational cross-sectional study 

Dear Dr. Ogrodzka-Ciechanowicz:

I'm pleased to inform you that your manuscript has been deemed suitable for publication in PLOS ONE. Congratulations! Your manuscript is now with our production department. 

Kind regards, 

on behalf of

Dr. Monika Błaszczyszyn 

Academic Editor

PLOS ONE